Workshop at the 6th Symposium on Advances in Approximate Bayesian Inference (non-archival), 2024 1–28

# Can a Confident Prior Replace a Cold Posterior?

**Martin Marek**                                                      MARTIN.MAREK.19@UCL.AC.UK
**Brooks Paige**                                                                B.PAIGE@UCL.AC.UK
*University College London*

**Pavel Izmailov**                                                                  PI390@NYU.EDU
*New York University*

## Abstract

Benchmark datasets used for image classification tend to have very low levels of label noise. When Bayesian neural networks are trained on these datasets, they often underfit, misrepresenting the aleatoric uncertainty of the data. A common solution is to cool the posterior, which improves fit to the training data but is challenging to interpret from a Bayesian perspective. We introduce a clipped version of the Dirichlet prior to control the aleatoric uncertainty of a Bayesian neural network, nearly matching the performance of cold posteriors within the standard Bayesian framework. We explain why the Dirichlet prior needs to be clipped in order to converge, and we derive the conditions under which it is numerically stable. We share our code at github.com/martin-marek/dirclip.

## 1. Introduction

When performing Bayesian classification, it is important to tune the model's level of aleatoric (data) uncertainty to correctly reflect the noise in the training data. For example, consider the binary classification problem in Figure 1 – if we believe that all of the plotted data points were labeled correctly, we would prefer a model that perfectly fits the data, using a complex decision boundary. In contrast, if we knew that the data labels were noisy, we might assume that the true decision boundary is actually simpler, and the two gray observations were mislabeled. Both of these decision boundaries provide reasonable descriptions of the data, and we can only choose between them based on our beliefs about the quality of the data labels.

In a regression setting, tuning the model's level of aleatoric uncertainty is a common practice, for example by modifying the kernel of a Gaussian process or by tuning the scale parameter of a Gaussian likelihood (Kapoor et al., 2022). In contrast, in a classification setting, we are forced to use the categorical likelihood, which has no tunable parameter to control the level of aleatoric uncertainty. The common solution in practice is to temper the posterior distribution, which softens/sharpens the likelihood. However, Wenzel et al. (2020) argued that tempering is problematic from a Bayesian perspective as it 1) deviates from the Bayes posterior, and 2) corresponds to a likelihood function that is not a valid distribution over classes. In this paper, we aim to show that we can match the results of posterior tempering within the standard Bayesian framework, by introducing a valid prior distribution that directly controls the aleatoric uncertainty of a Bayesian neural network (BNN).

Our paper is heavily inspired by and aims to extend the work of Kapoor et al. (2022). Kapoor et al. have provided a clear mechanistic explanation of how data augmentation

causes underfitting, as well as introducing the idea of using a Dirichlet prior to control the aleatoric uncertainty of a BNN. We show that the density of the Dirichlet prior that they proposed is unbounded, leading to an improper posterior, which caused the numerical instability in their experiments. As a result, Kapoor et al. (2022) used an approximation, which we show cannot be viewed as purely a prior modification. In this work, we propose a simple modification of the Dirichlet prior, that fixes the source of the instability, allowing us to match the results of the cold posterior with a pure modification of the prior, without the need for numerical approximations.

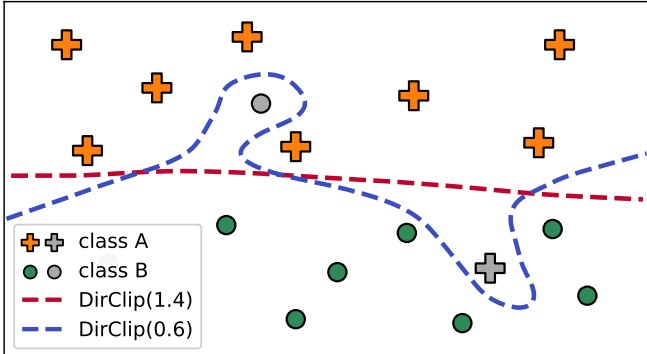

Figure 1: **Decision boundaries of a Bayesian neural network using the *DirClip* prior.** By varying the concentration parameter of the prior, we can control the model's aleatoric uncertainty, leading to different decision boundaries. The plotted decision boundaries were obtained using Hamiltonian Monte Carlo, using the dataset from Figure 1 of Kapoor et al. (2022).

## 2. Background

### 2.1. Bayesian neural networks

Bayesian neural networks are an exciting framework for both understanding and training neural networks that are more reliable (Mackay, 1992; Neal, 2012; Wilson and Izmailov, 2020). A trained BNN is fully defined by its posterior distribution over parameters. Let's denote the model parameters $\boldsymbol{\theta}$, inputs (e.g., images) $\mathbf{X}$ and labels $\mathbf{Y}$. Then the posterior is proportional to a prior times a likelihood:

$$\underbrace{p(\boldsymbol{\theta}|\mathbf{X}, \mathbf{Y})}_{\text{posterior}} \propto \underbrace{p(\boldsymbol{\theta}|\mathbf{X})}_{\text{prior}} \underbrace{p(\mathbf{Y}|\boldsymbol{\theta}, \mathbf{X})}_{\text{likelihood}}. \tag{1}$$

### 2.2. Cold posteriors

A cold posterior is achieved by exponentiating the posterior to $1/T$, where $T < 1$. Since the posterior factorizes into a prior and a likelihood, a cold posterior can be seen as a combination of a cold likelihood with a cold prior:

$$p(\boldsymbol{\theta}|\mathbf{X}, \mathbf{Y})^{1/T} \propto p(\boldsymbol{\theta}|\mathbf{X})^{1/T} p(\mathbf{Y}|\boldsymbol{\theta}, \mathbf{X})^{1/T}. \tag{2}$$

Wenzel et al. (2020) showed that the standard posteriors corresponding to $T = 1$ lead to poor performance in image classification, while cold posteriors with $T < 1$ provide much better performance. They named this observation the *cold posterior effect* (CPE). The cold posterior effect can seem problematic from a theoretical point of view, because a cold posterior corresponds to using a cold categorical likelihood, which is not a valid distribution over classes (Wenzel et al., 2020).

Several works have attempted to explain the CPE or propose methods that would eliminate it. Izmailov et al. (2021b) repeated the experiments of Wenzel et al. (2020) but with data augmentation turned off, which entirely eliminated the cold posterior effect. Kapoor et al. (2022) showed that naively implementing data augmentation results in undercounting the training data, which softens the likelihood and directly leads to underfitting. One way to counteract this effect is to use a cold posterior, which sharpens the likelihood and directly compensates for the undercounting. Nabarro et al. (2022) proposed a principled version of data augmentation and showed that the cold posterior effect remains in their model, suggesting that other factors can contribute to the cold posterior effect. Noci et al. (2021) trained a BNN on subsets of a dataset of varying size and observed that as the dataset size decreased, the strength of the cold posterior effect increased. This observation implies that the Normal prior is misspecified, especially when the dataset is small or the model is large. Based on the aforementioned work, we conclude that the cold posterior effect has multiple possible causes, although they all seem to stem from underfitting (Zhang et al., 2023) or overestimating the aleatoric uncertainty of the training data (Adlam et al., 2020). In order to eliminate the cold posterior effect, we therefore search for a solution in the form of an improved prior distribution, to improve fit to the training data.

## 3. Dirichlet prior

In a Bayesian classification setting, if we wish to control a model's level of aleatoric uncertainty, the standard approach is to use a Dirichlet prior, which can bias the posterior to either have lower or higher confidence. More formally, the Dirichlet distribution is a distribution over class probabilities. Let's denote a model's predicted class probabilities as $\hat{\mathbf{y}} = (\hat{y}_1, \hat{y}_2 \ldots \hat{y}_K)$, where $K$ is the number of classes. The Dirichlet prior assigns a probability density to any set of predictions $\hat{\mathbf{y}}$:

$$\log p(\hat{\mathbf{y}}) \stackrel{c}{=} \sum_{k=1}^{K} (\alpha - 1) \log \hat{y}_k. \tag{3}$$

Kapoor et al. (2022) have observed that directly using the Dirichlet prior as a prior over parameters $p(\boldsymbol{\theta}) = p(\hat{\mathbf{y}})$ results in divergence. The issue is that the Dirichlet prior is a distribution over model *predictions*, whereas the BNN prior as defined in Eq. (1) is a distribution over model *parameters*. So how can we translate a distribution over predictions $p(\hat{\mathbf{y}})$ into a distribution over parameters $p(\boldsymbol{\theta})$? In general, this is a difficult problem with no simple solution; we discuss this further in Appendix F.1. In the rest of this paper, we search for a prior that works well when applied directly over model parameters.

To understand why the Dirichlet prior diverges when applied over model parameters, we plot its probability density function (PDF) on the left side of Figure 2. Notice that the

probability density diverges to $\infty$ as $\hat{y}$ approaches either 0 or 1. For a probability distribution to be valid, its PDF needs to integrate to one but the PDF does not necessarily need to be bounded. Indeed, if we treat the Dirichlet prior as a distribution over the domain $\hat{y} \in [0, 1]$, the PDF integrates to one over this domain, and it is, therefore, a valid distribution. The fact that the PDF is unbounded becomes an issue when we treat the PDF as a distribution over parameters because the domain of model parameters is $\boldsymbol{\theta} \in (-\infty, \infty)$. If we attempt to integrate the PDF over the domain of model parameters, we get an integral that diverges, meaning that the distribution is invalid. Even worse, the *posterior* distribution that results from this prior is also invalid and hence impossible to sample. We prove the divergence of the Dirichlet prior more formally in Appendix F.2.

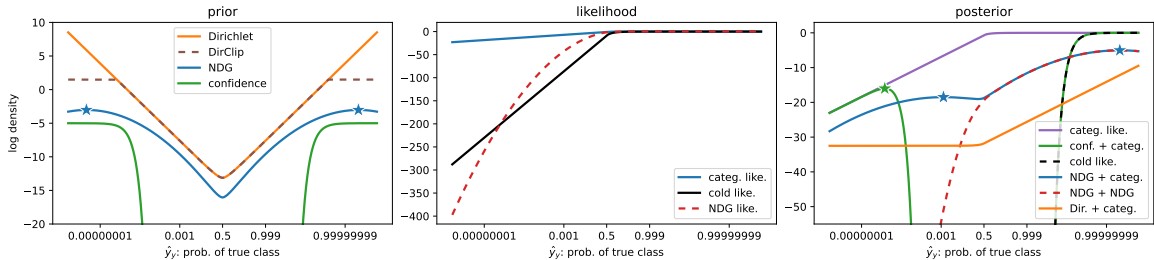

Figure 2: **Slices of various prior, likelihood, and posterior distributions.** For each distribution, we assume that there are only two classes and we vary the predicted probability of the true class on the x-axis. Since the prior has no notion of the "true" class, it is symmetric. Note that the x-axis is non-linear to better show the tail behavior of each distribution. Notably, the NDG prior peaks at a very small (and large) value of predicted probability, which would not be visible on a linear scale. The blue and green stars in the left and right plots show local maxima.

## 4. Noisy Dirichlet Gaussian

To fix the divergence of the Dirichlet prior, Kapoor et al. (2022) decided to approximate the *product* (posterior) of the Dirichlet prior with a categorical likelihood. They name this method *Noisy Dirichlet Gaussian (NDG)*. In their experiments, NDG closely matches the performance of a cold posterior.

However, we show in Appendix D that the NDG posterior approximation corresponds to using a valid prior *combined* with a quadratic likelihood term. This raises the question: does the NDG posterior work so well because of its implied prior distribution or because of the implied quadratic likelihood? To answer this, we perform a simple experiment. First, we train a model that uses the NDG prior together with the categorical likelihood. Second, we train a model that uses the NDG quadratic likelihood. Figure 3 shows that neither the NDG prior nor the NDG likelihood on their own can match the test accuracy of a cold posterior.

The reason that the NDG prior works at all (unlike a Dirichlet prior) is that its density is bounded. This can be seen in the left plot of Figure 2: whereas the Dirichlet prior diverges toward each tail of the distribution, the NDG prior density is bounded by a local maximum

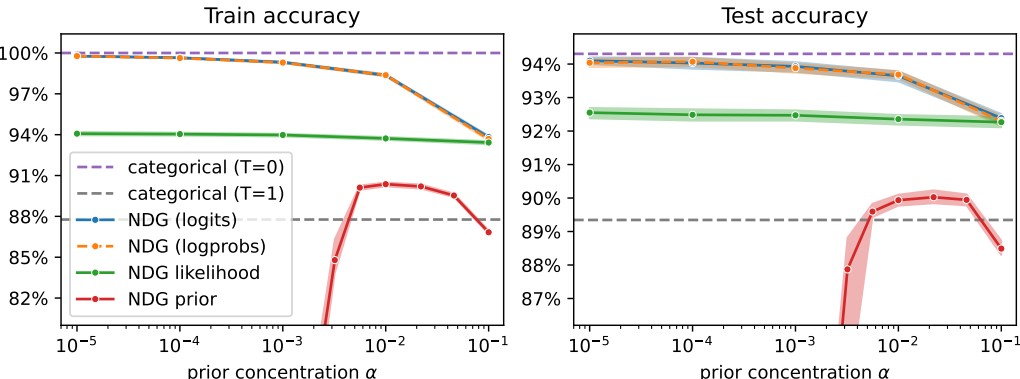

Figure 3: **Factorized NDG.** This figure shows the accuracy of ResNet20 on CIFAR-10 with data augmentation for various BNN posteriors. Each posterior consists of a $\mathcal{N}(0, 0.1^2)$ prior over model parameters, a (modified) likelihood, and optionally an additional prior term over predictions. The model using the standard categorical ($T$=1) likelihood provides a simple baseline. The NDG posterior models defined over logits and log-probabilities[2] both reach the same test accuracy, on par with a cold posterior. In contrast, the NDG prior and likelihood on their own do not match the performance of a cold posterior. Note that the training accuracy was evaluated on posterior *samples*, whereas the test accuracy was evaluated on the posterior *ensemble*.

at each tail of the distribution. If we combine any bounded prior distribution with a Normal prior (with an arbitrarily large scale), we get a valid prior distribution over parameters.[3]

## 5. Clipped Dirichlet prior

The NDG prior provides an approximation to the Dirichlet prior that fixes its divergence. However, there exists a simpler method to fix the divergence of the Dirichlet prior: we can clip each log-probability at some small value $v$ (for example, $v = -10$):

$$\log p(\hat{\mathbf{y}}) \overset{c}{=} \sum_{k=1}^{K} (\alpha - 1) \max(\log \hat{y}_k, v). \tag{4}$$

Since the log-density of the Dirichlet prior is proportional to the sum of per-class log-probabilities, clipping each log-probability bounds the prior density. We call this prior *DirClip*, as in "Dirichlet Clipped". Note that a similar modification of the Dirichlet distribution has been proposed to allow the concentration parameter $\alpha$ to become negative,

---

2. We discuss the difference between the NDG posterior defined over logits and log-probabilities in Appendix D.

3. Since the Normal prior is a proper prior, even if we combine it with an improper prior distribution, the joint prior will be proper. In practice, there is little difference between using the NDG prior on its own vs. combining it with a large-scale Normal prior; it is simply worth realizing that the latter prior is guaranteed to be proper and therefore result in a proper posterior.

although it was never applied as a prior over model parameters (Tu, 2016). The DirClip prior is visualized in the left plot of Figure 2: it is identical to the Dirichlet prior for log-probabilities between the clipping value $v$ and it stays at the clipping value otherwise. We define the DirClip posterior as follows:

$$\underbrace{p(\boldsymbol{\theta}|\mathbf{X},\mathbf{Y})}_{\substack{\text{DirClip}\\\text{posterior}}} \propto \underbrace{p(\boldsymbol{\theta})}_{\substack{\text{Normal}\\\text{prior}}} \prod_{i=1}^{N} \overbrace{\underbrace{p(\hat{\mathbf{y}}^{(i)})}_{\substack{\text{DirClip}\\\text{prior}}} \underbrace{\hat{y}_y^{(i)}}_{\substack{\text{categorical}\\\text{likelihood}}}}^{\text{evaluated per observation}}. \tag{5}$$

**Results.** We show in Figure 1 that the DirClip prior can be used to control the level of aleatoric uncertainty on a toy binary classification dataset. In Figure 4, we show that the DirClip prior can be used to control the training accuracy of a ResNet20 trained on CIFAR-10. When $\alpha = 1$, the DirClip prior is equivalent to a uniform prior, and the model underfits. By increasing the prior concentration (reducing $\alpha$), we can increase training accuracy from 88% to 99% and test accuracy from 89% to almost 94%. However, notice in the figure that the model trained from random initialization only converges when $\alpha > 0.8$. When $\alpha < 0.8$, the accuracy suddenly drops all the way down to 10% (corresponding to random guessing). This sudden drop in accuracy is not a fundamental property of the DirClip prior; instead, it can be entirely attributed to the challenges of gradient-based sampling algorithms. We explain this behavior in Appendix C.1.

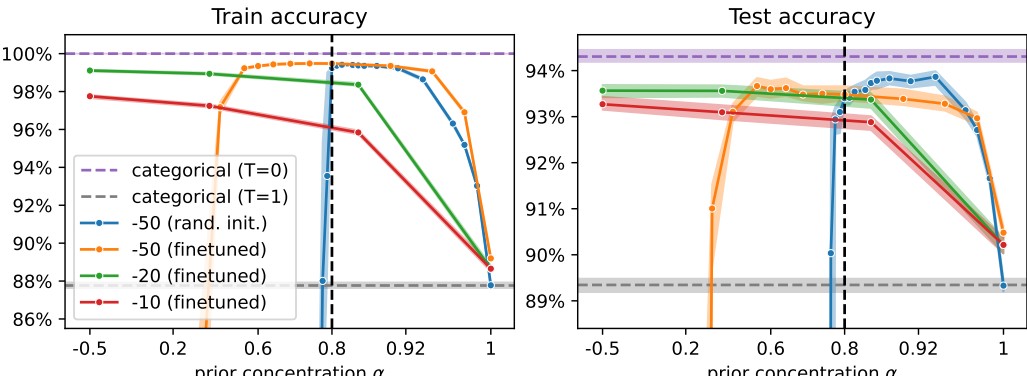

Figure 4: **DirClip accuracy**. This figure shows the accuracy of ResNet20 on CIFAR-10 with data augmentation for various BNN posteriors. Each solid line corresponds to a different clipping value of the DirClip prior (printed in the legend). The blue line shows DirClip posteriors sampled from random initialization; all other DirClip posteriors are fine-tuned from a checkpoint with 100% training accuracy. Note that the training accuracy was evaluated on posterior *samples*, whereas the test accuracy was evaluated on the posterior *ensemble*.

**Summary.** We acknowledge that cold posteriors are significantly easier to sample than the *DirClip* prior, and therefore remain the practical solution when the model is misspecified (Wilson and Izmailov, 2020). The goal of this paper was to show that tempering is not *necessary* to achieve high accuracy on CIFAR-10; we can instead use a prior that enforces high confidence on the training data and is entirely consistent with the Bayesian framework.

## Acknowledgments

This research was supported by Google's TPU Research Cloud (TRC) program: https://sites.research.google/trc/.

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

### Appendix outline

- In Appendix A, we show that a Normal prior can simultaneously obtain high prior confidence and low posterior confidence.

- In Appendix B, we visualize the effect of posterior tempering and compare the accuracy and likelihood achieved by all models discussed in this paper.

- In Appendix C, we explain why the DirClip prior is numerically unstable and show that it reaches its clipping value, which results in low test likelihood when the posterior distribution is approximated using a small number of samples.

- In Appendix D, we factorize the NDG posterior into a prior and a likelihood.

- In Appendix E, we explain that the DirClip prior is actually a prior on small log-probabilities, rather than high confidence. Using this observation, we propose a prior that directly encourages high confidence.

- In Appendix F, we explain why correctly sampling from a prior over model outputs requires a "change of variables" term; we prove that the Dirichlet prior diverges if this term is omitted; and we prove that the DirClip prior is a valid distribution.

- In Appendix G, we derive the direction of a gradient step along the Dirichlet posterior and show when it increases the probability of the observed class.

- In Appendix H, we provide implementation details for all of our experiments.

### Appendix A. Confidence of a Normal prior

Most prior works studying Bayesian neural networks used isotropic Normal priors (Wenzel et al., 2020; Noci et al., 2021; Sharma et al., 2023) or other vague distributions over parameters, such as a Mixture of Gaussians, logistic distribution (Izmailov et al., 2021b), Laplace distribution, Student's t-distribution (Fortuin et al., 2021), or a correlated Normal distribution (Izmailov et al., 2021a; Fortuin et al., 2021). We show that these distributions can have high *prior* confidence, while simultaneously having low *posterior* confidence. This implies that the link between prior and posterior confidence is not trivial, and using a confident prior does not guarantee high posterior confidence.

We performed a simple experiment where we varied the scale of the Normal prior, sampled ResNet20 parameters from the prior, and evaluated its predictions on the training set of CIFAR-10. The dashed line in Figure 5 shows the average confidence of prior samples as a function of the prior scale. Intuitively, as the prior scale tends to zero, the model parameters tend to zero, causing logits to approach zero, thereby inducing uniform predictions over class probabilities. Conversely, as the prior scale increases, model parameters grow in magnitude, scaling up the logits. When the scale of logits is large, the absolute differences between the logits grow, leading to high confidence.

Observe in Figure 5 that prior confidence does not trivially translate into posterior confidence. Each point corresponds to a single posterior distribution, with color representing the posterior temperature. All untempered $(T = 1)$ posteriors have low confidence, irrespective

of the prior scale. When a cold temperature is used, it is possible to obtain a model that has prior confidence near 10% (the lowest possible) *and* posterior confidence of 99%.

These results show that if we want to train a BNN that has high posterior confidence at $T = 1$, it is *not* sufficient to use a prior whose samples are confident. One way to increase the posterior confidence further is to use a prior that will directly assign a high probability density to models with high confidence and a low probability density to models with low confidence. This approach requires a *functional* prior, i.e. a prior distribution that is defined over functions (model predictions), rather than model parameters.[4]

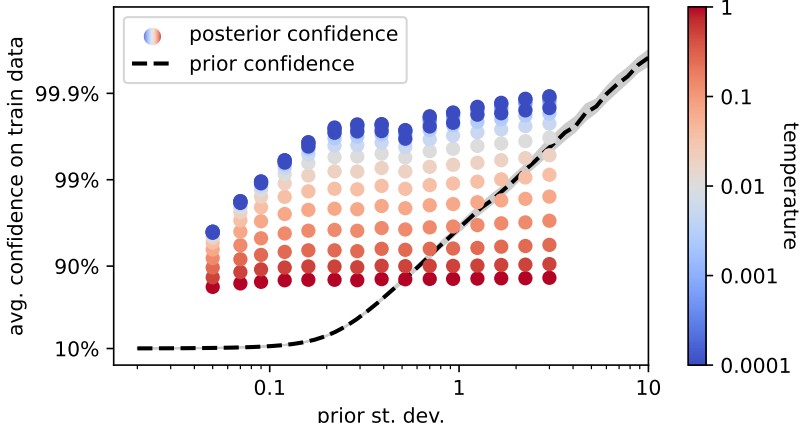

Figure 5: **Confidence of ResNet20 trained on CIFAR-10 with a Normal prior.** The dashed line shows the average confidence of prior samples as a function of the prior scale (standard deviation). The relationship is one-to-one: the prior scale exactly determines prior confidence. Conversely, the prior scale has almost no effect on posterior confidence—each scatter point corresponds to a single trained model. Here, the intuition that "prior confidence translates into posterior confidence" fails. Instead, the posterior confidence depends mostly on the posterior temperature, visualized using the colorbar on the right.

## Appendix B. Model comparison

To visualize the effect of posterior tempering on the training accuracy, test accuracy, and parameter norm, we sampled 16 different posterior temperatures across 15 different prior scales—the results are shown in Figure 6.

In Figure 7, we compare the training accuracy, test accuracy, and test likelihood of various posterior distributions. Observe in the left plot that the training accuracy almost perfectly predicts the test accuracy. A possible explanation behind this effect is that most of the posteriors underfit the training data, therefore overestimating the aleatoric uncertainty

---

4. In theory, every distribution over model parameters corresponds to some distribution over functions, and vice versa. We argue that in order to design a confidence-inducing prior distribution, it is *helpful* (although not strictly necessary) to directly think about the prior in function space.

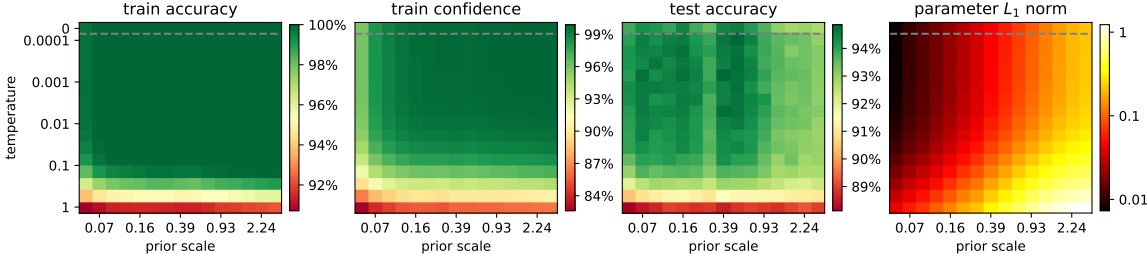

Figure 6: **Effect of posterior temperature and Normal prior scale on a Bayesian neural network.** The model is a ResNet20 trained on CIFAR-10 with data augmentation turned on. When $T = 1$, the augmented data is undercounted, resulting in underfitting (low training accuracy). As temperature decreases, the fit to training data improves, increasing test accuracy. At the same time, as temperature decreases, the norm of the model parameters decreases, but only up to a point. With a decreasing temperature, the cold posterior approaches a deep ensemble, which is obtained by setting the temperature to exactly zero.

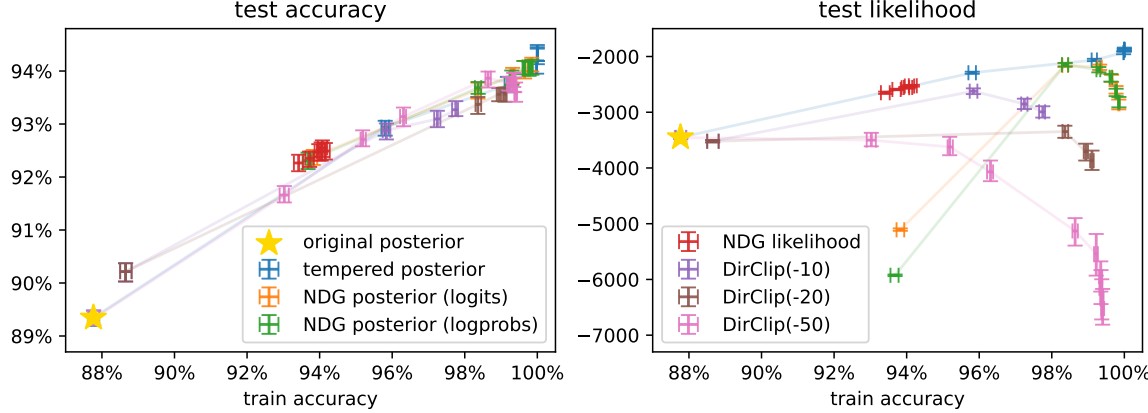

Figure 7: **Comparison of various methods for controlling aleatoric uncertainty.** The model is a ResNet20 trained on CIFAR-10 with data augmentation turned on. The *original posterior* consists of a $\mathcal{N}(0, 0.1^2)$ prior over model parameters and the categorical likelihood; the *tempered posterior* uses the same prior and likelihood, but the posterior temperature is varied. The *NDG* models use a $\mathcal{N}(0, 0.1^2)$ prior over parameters, combined with either the full NDG posterior or only the quadratic NDG likelihood on its own. Lastly, the DirClip models use varying concentration parameters $\alpha$ with the clipping value shown in the plot legend.

in the labels (Adlam et al., 2020). In order to correctly represent the aleatoric uncertainty

in the data labels, the model needs to reach near 100% training accuracy. However, the behavior of the test likelihood is more complex—we discuss this in Appendix C.3.

## Appendix C. Further discussion of DirClip prior

### C.1. Training stability

Figure 8 shows the paths that 50 randomly sampled particles would take along the domain of various probability distributions under gradient descent in logit space. Notice that the gradient of the categorical likelihood always points toward the top corner (correct class). In contrast, the Dirichlet prior has no notion of a "correct" class, and its gradient simply points toward whichever corner (class) is the closest. When the Dirichlet prior is combined with the categorical likelihood, the combined gradient does not necessarily point toward the correct class. For example, in Figure 8c, around 30% of uniformly-sampled particles would converge to either of the bottom corners under gradient descent. However, if we directly sampled the Dirichlet posterior, only 1% of the samples would appear in the bottom half of the plot. Therefore, this behavior of particles converging to the wrong class is purely an optimization problem, rather than a statistical property of the Dirichlet distribution.

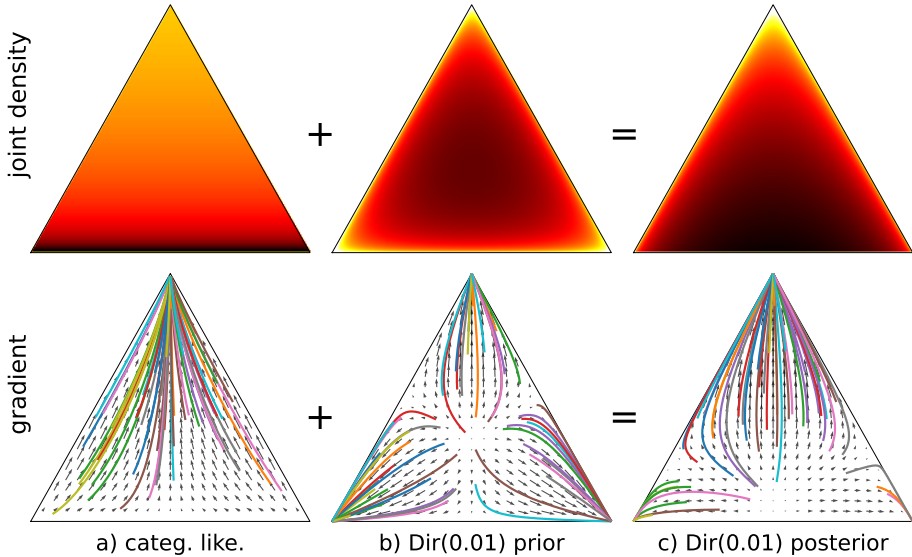

Figure 8: **Top:** probability density of a) categorical likelihood, b) Dirichlet prior, c) Dirichlet prior combined with categorical likelihood. Each distribution is defined over three classes. The top corner corresponds to the correct class and the two bottom corners correspond to the two other classes. **Bottom:** vector field shows the direction and magnitude of gradients computed in logit space and reprojected back to the probability simplex. The colored lines show the trajectories of 50 randomly sampled particles under this gradient field.

The same behavior translates to a Bayesian neural network optimized (sampled) using gradient-based methods. If the neural network predicts a wrong class at initialization, optimizing the parameters of the neural network may further *increase* the model's predicted probability for that class. The model does not necessarily converge to predicting the correct class.

In the case of the Dirichlet posterior, we can analytically derive whether a gradient step will increase or decrease the probability of the true class (full derivation is provided in Appendix G). Intuitively, if a model's prediction is initially close to the true class, then gradient descent will increase the probability of that class. However, when the prediction is initially close to a wrong class, the gradient *may* point toward the wrong class. Figure 9 visualizes this behavior using a phase diagram. If a neural network is randomly initialized and $\alpha < 0.8$, most of its predictions will fall inside the diverging phase, meaning those predictions may not converge toward the true class. There are two ways to fix this: 1) use $\alpha > 0.8$; or 2) initialize the neural network such that all of its predictions fall inside the converging phase. In practical terms, this means fine-tuning the neural network from a checkpoint that has 100% training accuracy. In Figure 4, we show that the fine-tuning strategy works for $\alpha < 0.8$.[5] For further discussion of the DirClip prior, please see Appendix C.

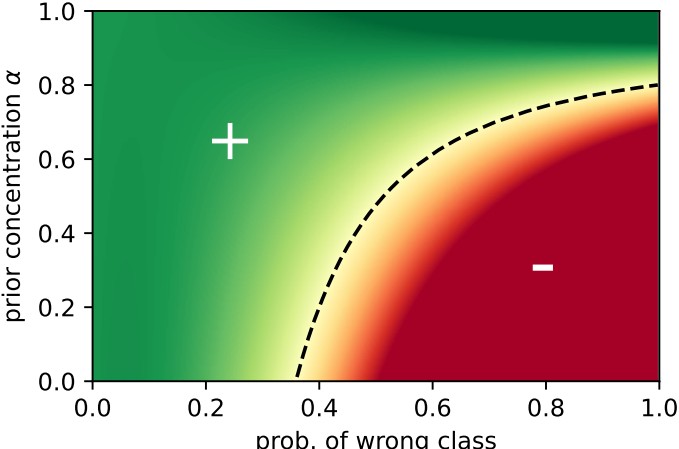

Figure 9: *Does a gradient step on the Dirichlet posterior increase the probability of the correct class?* Assume that there are 10 classes. The x-axis shows the probability of a *wrong* class; the other 9 classes (one of which is the correct class) have the same probability. The image color shows the change in the probability of the correct class under a gradient step. Green means that the probability of the correct class will increase; red means that it will decrease. Notice that for $\alpha > 0.8$, the probability of the correct class always increases. For $\alpha < 0.8$, the probability of the correct class may increase or decrease, depending on the x-axis value.

---

5. fine-tuning is necessary but not sufficient for convergence with $\alpha < 0.8$. In particular, we found that smaller values of $\alpha$ require increasingly smaller learning rates. The stability phase diagram in Figure 9 only holds for an infinitely small learning rate.

## C.2. Clipping value is reached

In Figure 4, we show that the DirClip prior achieves higher training accuracy when the clipping value −50 is used, compared to the clipping value −10. This might seem surprising because a log-probability of −10 is already a very small value. The reason the clipping value affects the prior behavior is that the clipping value *is* reached on most predictions. Observe in Figure 10 that almost 90% of the predicted log-probabilities of DirClip(−50) posterior samples are smaller than its clipping value of −50. Similarly, almost 90% of the predicted log-probabilities of DirClip(−10) posterior samples are smaller than its clipping value of −10. Note that CIFAR-10 has 10 classes. Therefore, these models have converged approximately to predicting the clipping value for each wrong class and predicting a log-probability close to 0 for the true class.

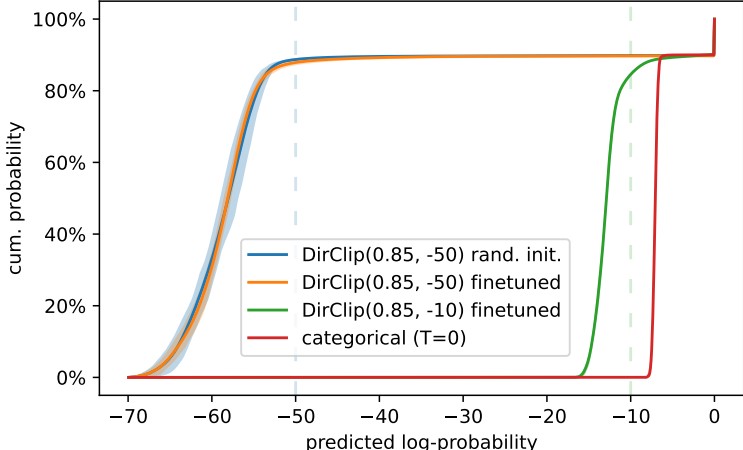

Figure 10: **CDF of predicted log-probabilities on training data.** We sampled various ResNet20 posteriors on CIFAR-10. Afterwards, we independently evaluated the predictions of each posterior sample on the training data, concatenating all predicted log-probabilities across classes and posterior samples.

For comparison, notice that cold posterior samples achieve perfect training accuracy (Fig. 4) without predicting extremely small log-probabilities. This relates to the issue discussed in Appendix E: the Dirichlet prior enforces small probabilities rather than directly enforcing high confidence (which is not the same thing). In particular, predicting extremely small log-probabilities negatively affects the test likelihood of the DirClip prior.

## C.3. Low likelihood

In Figure 7, we compare the accuracy and likelihood of various posterior distributions. Note that as we increase the concentration of the DirClip(−50) prior, the train and test accuracy increase while the test likelihood *decreases*. The opposite is true for the cold posterior: its test accuracy increases *together* with the test likelihood.

The low likelihood of the DirClip(−50) prior can be directly attributed to the small log-probabilities that it predicts. It not only predicts class log-probabilities of −50 on the training data (as shown in Figure 10) but also on the test data, sometimes predicting a log-probability of −50 for the *correct class*. Such a prediction incurs an extremely low likelihood. In Figure 11, we show that the DirClip posterior samples misclassify almost 10% of test images with log-probability of less than −10, while samples from the cold posterior have exactly *zero* such overconfident misclassifications.

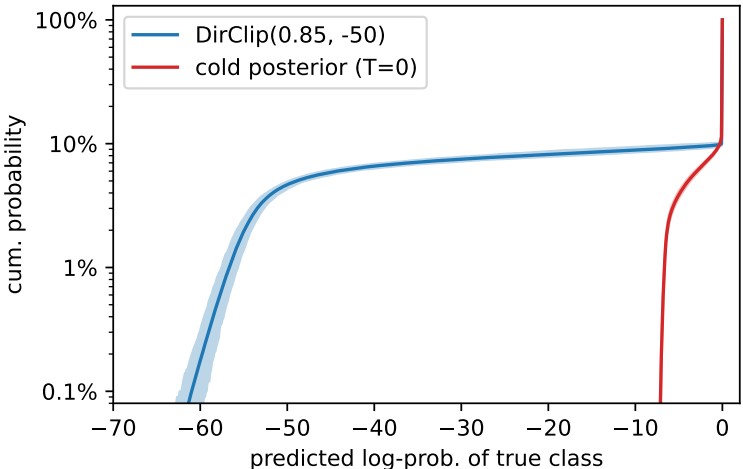

Figure 11: **CDF of predicted log-probabilities for the true class on test data.** We sampled the DirClip and cold ResNet20 posteriors on CIFAR-10. Afterwards, we independently evaluated the predictions of each posterior sample on the test data, concatenating all predicted log-probabilities for the true class.

Intuitively, a BNN can misclassify an image with extremely high confidence only if all posterior samples assign an extremely low probability to the true label. Most of our experiments only used 8 (completely independent) posterior samples, so this happens often. If we instead used more posterior samples, it is more likely that at least one of the posterior samples would assign a higher probability to the true class, which would significantly improve the test likelihood of the DirClip prior. In Figure 20, we show that as we increase the number of posterior samples, the likelihood of the DirClip prior indeed significantly improves. Whereas the likelihood of the cold posterior converges after only ∼20 posterior samples, the DirClip posterior requires 200 samples to reach a similar likelihood value.

On the one hand, this implies that the low likelihood of the DirClip posterior in Figure 7 is an artifact of our approximate inference pipeline, rather than a fundamental property of the true DirClip posterior distribution. On the other hand, the fact that the DirClip posterior requires an order of magnitude more posterior samples to converge (compared to a cold posterior) is a critical difference for any practitioner. We discuss the (high) computational cost of our experiments in Appendix H.

### C.4. fine-tuning has converged

In Figure 4, we compared the accuracy of both randomly initialized and fine-tuned DirClip models. Given that the fine-tuned models were initialized from a checkpoint with 100% train accuracy, how can we know that the fine-tuning has converged? One check that we performed was comparing the CDF of predicted log-probabilities, as shown in Figure 10. At initialization, the fine-tuned DirClip model had a similar CDF to the cold posterior. However, after fine-tuning, the CDF converged to the CDF of the randomly initialized DirClip model, suggesting that the fine-tuning has converged.

## Appendix D. Factorization of NDG posterior

Kapoor et al. (2022) approximated the product of a Dirichlet($\alpha$) prior with the categorical likelihood using a Gaussian distribution over the model's predicted log-probabilities. They named this distribution *Noisy Dirichlet Gaussian (NDG)*. We visualize the NDG distribution in Figure 12 and formally define its density function in Eq. (6).

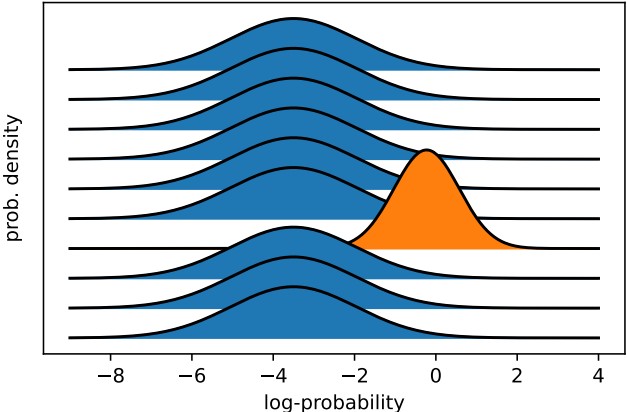

Figure 12: **Visualization of the Noisy Dirichlet Gaussian (NDG) distribution over model's predicted log-probabilities.** The predicted log-probability for each class is assigned an independent Normal distribution. The log-probability corresponding to the correct class has a shifted mean and reduced standard deviation; all other log-probabilities have the same distribution.

$$
\begin{aligned}
\text{NDG}(\log \hat{\mathbf{y}} | \tilde{\boldsymbol{\alpha}})_k &\sim \mathcal{N}(\log \hat{y}_k | \mu_k, \sigma_k^2), \text{ where} \\
\tilde{\alpha}_k &= \alpha + \mathbb{1}(k = y) \\
\sigma_k &= \log(\tilde{\alpha}_k^{-1} + 1) \\
\mu_k &= \log(\tilde{\alpha}_k) - \log(\tilde{\alpha}_y) + (\sigma_y^2 - \sigma_k^2)/2
\end{aligned}
\tag{6}
$$

Technically, the original NDG model is a Gaussian distribution over predicted logits rather than log-probabilities. However, logits and log-probabilities are always equivalent up

to a constant. Therefore, the only difference between defining the distribution over logits and log-probabilities is that using logits removes one degree of freedom. We experimentally verify that using NDG over logits and log-probabilities is equivalent in Figure 3.[6]

Unfortunately, it turns out that the NDG approximation effectively uses a quadratic likelihood term. Recall that NDG is not just an approximate prior; instead, it jointly approximates the prior *together* with the likelihood. We factorize the $\text{NDG}(\tilde{\boldsymbol{\alpha}})$ posterior in Eq. (7), showing that it is equal to a combination of a reparameterized $\text{NDG}(\boldsymbol{\alpha})$ prior and a quadratic likelihood term. The implied prior and likelihood functions are plotted in the left and middle plots of Figure 2.

$$
\begin{aligned}
\text{let } \log \hat{y}_k &\sim \mathcal{N}(\mu_k, \sigma_k^2) \text{ where} \\
\mu_k &= \mu_1 \text{ if } k = y \text{ else } \mu_0 \\
\sigma_k &= \sigma_1 \text{ if } k = y \text{ else } \sigma_0 \\
\log p(\log \hat{\mathbf{y}}) &\stackrel{c}{=} -\frac{1}{2} \sum_k \frac{(\log \hat{y}_k - \mu_k)^2}{\sigma_k^2} \\
&\stackrel{c}{=} -\frac{1}{2} \left( \sum_k \frac{(\log \hat{y}_k - \mu_0)^2}{\sigma_0^2} \right) + \\
&\quad \left( \frac{\sigma_0^2 \mu_1 - \sigma_1^2 \mu_0}{\sigma_0^2 \sigma_1^2} \log \hat{y}_y + \frac{\sigma_1^2 - \sigma_0^2}{2\sigma_0^2 \sigma_1^2} \log \hat{y}_y^2 \right)
\end{aligned}
\tag{7}
$$

## Appendix E. Confidence prior

While the DirClip prior introduced in Section 5 *can* successfully control a model's level of aleatoric uncertainty, it achieves this in a very indirect way. To see this, consider the probability density that the DirClip(0.01) prior assigns to the following two predictions:

$$
\begin{aligned}
p((99\%, 0.5\%, 0.5\%)) &\approx 3.6 \\
p((49.99999\%, 49.99999\%, 0.00002\%)) &\approx 1102.
\end{aligned}
\tag{8}
$$

The probability density assigned to the second prediction is roughly *300-times* higher even though it only has half the confidence of the first prediction ($\sim$50% vs. 99%). The issue is that the DirClip prior (and the Dirichlet distribution in general) assigns a high density to predictions with *small probabilities* rather than predictions with high confidence. This can be seen directly by looking at the Dirichlet PDF in Eq. (3): when $\alpha < 1$, the density is high when any of the log-probabilities is extremely small. In particular, high density of the Dirichlet distribution does not directly imply high confidence.

Instead of using a prior that encourages small log-probabilities, we can design a prior that directly enforces high prediction confidence. We can simply set the density of the prior proportional to the confidence (i.e. the maximum predicted probability):

$$
p(\hat{\boldsymbol{y}}) \propto \left( \max_k \hat{y}_k \right)^{1/T-1}.
\tag{9}
$$

6. When defining the distribution over log-probabilities, we shifted $\mu_k$ by a constant s.t. $\mu_y = 0$.

We intentionally parameterized this "confidence prior" using $T$, so that when the predicted class matches the true class (which is guaranteed to be true when $\hat{y}_y \geq 0.5$), the density of the confidence prior combined with categorical likelihood is equal to the density of a cold categorical likelihood:

$$\operatorname*{argmax}_k \hat{y}_k = y \implies p(\hat{\boldsymbol{y}}) \cdot \hat{y}_y = \hat{y}_y^{1/T}. \tag{10}$$

On the left side of Figure 13, we plot the density of tempered categorical likelihood, which also provides a lower bound on the product of the confidence prior with untempered categorical likelihood. On the right side, we plot the upper bound. Notice that as the temperature approaches zero, both densities converge to the Dirac delta measure concentrated at 1. We prove this convergence in Appendix E.1.

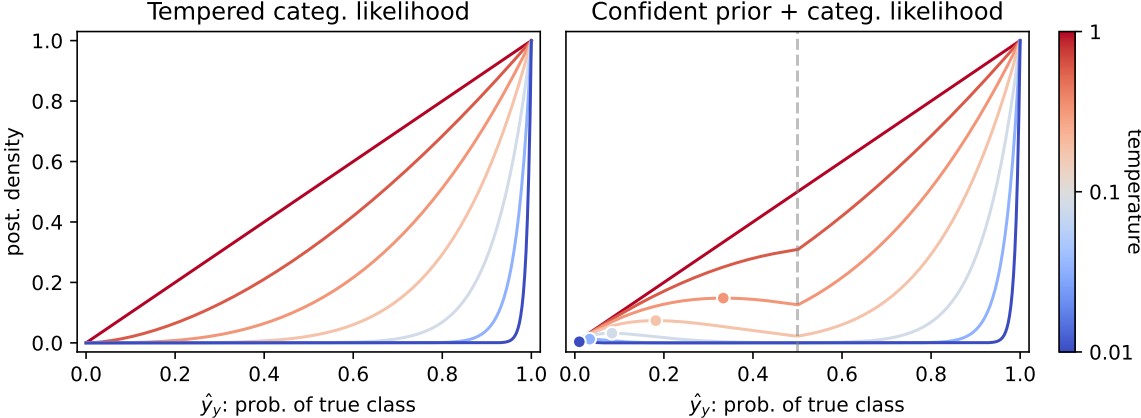

Figure 13: **Left:** Density of the categorical likelihood tempered to various temperatures (shown by the colorbar on the right). A cold likelihood ($T < 1$) is "sharp" and penalizes wrong predictions heavily. The red line shows the untempered likelihood ($T = 1$). **Right:** An upper bound on the product of a confident prior (parameterized using temperature) with the untempered categorical likelihood. Note that for $\hat{y}_y > 0.5$, the two densities are identical.

Recall from Eq. (2) that when a Normal prior and categorical likelihood is used, a cold posterior is equivalent to rescaling the prior and tempering the likelihood. However, it is also possible to approximate the cold posterior with a "confidence posterior". The "confidence posterior" consists of a rescaled Normal prior, confidence prior, and *untempered* categorical likelihood:

$$\underbrace{p(\boldsymbol{\theta}|\mathbf{X},\mathbf{Y})^{\frac{1}{T}}}_{\text{cold posterior}} \approx \underbrace{p(\boldsymbol{\theta}|T\sigma^2)}_{\substack{\text{rescaled} \\ \text{Normal prior}}} \prod_{i=1}^{N} \overbrace{\underbrace{\left(\max_k \hat{y}_k^{(i)}\right)^{\frac{1}{T}-1}}_{\text{confidence prior}} \underbrace{\hat{y}_y^{(i)}}_{\substack{\text{categorical} \\ \text{likelihood}}}}^{\text{approximates cold likelihood}}. \tag{11}$$

In fact, if the posterior mode reaches 100% accuracy, then the confidence posterior and cold posterior have the same Hessian, implying that the two distributions have an identical Laplace approximation. This follows directly from Eq. (10).

In Figure 14, we visualize various posterior distributions. Notice that the *categorical likelihood* is relatively diffuse, assigning high probability density to models that deviate from the SGD solutions. In contrast, both the *cold likelihood* and the *confidence posterior* are very "sharp"—they are concentrated closely around the three trained models. Lastly, the DirClip posterior shows a surprising behavior, assigning the highest probability density to a model that only has a 10% training accuracy. The reason is that the top region of the plot happens to minimize the predicted log-probabilities and therefore maximize the DirClip prior density.

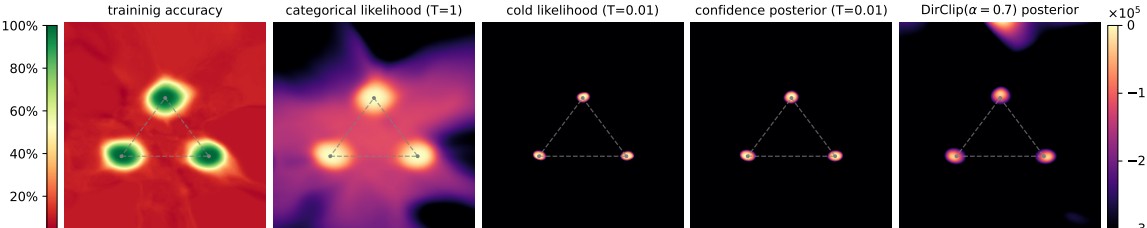

Figure 14: **Posterior landscapes.** Each plot shows a gray triangle whose vertices correspond to different ResNet20 models trained on CIFAR-10 using SGD. We evaluate the training accuracy and various distributions along the plane defined by the parameters of the three trained models. The colorbar on the left shows the accuracy scale and the colorbar on the right shows the distribution log-density.

Unfortunately, it is not possible to sample the confidence posterior using SGHMC or related optimization methods. The issue is that, unlike a cold likelihood, the confidence prior combined with categorical likelihood is riddled with local maxima. In Figure 13, each dot in the right plot shows a local maximum, using a linear y-axis scale. While these maxima may not seem too steep, they get steeper with decreasing temperature. The right plot in Figure 2 shows the same local maximum on a logarithmic scale for a lower temperature parameter $\left(T = 3 \cdot 10^{-7}\right)$. In theory, if we were sampling points uniformly under the "confidence posterior" curve in Figure 2, more than 99.9999% of the sampled points would end up on the right. However, local optimization methods such as SGD and SGHMC might get stuck in the local minimum on the left. Depending on how these local optimization algorithms are initialized, up to 50% of the sampled points might end up on the left.[7]

### E.1. Proof that confidence prior converges to a cold likelihood

In this section, we show that both a cold likelihood and the confidence prior combined with untempered categorical likelihood converge to a Dirac delta measure as $T \to 0$. We prove this by deriving lower and upper bounds on the product of the confidence prior with the categorical likelihood and showing that they both converge to the same distribution.

---

7. Sampling algorithms like HMC generate unbiased but correlated posterior samples. In theory, the posterior approximation gets more accurate the longer we run the sampling algorithm. In practice, however, it might be infeasible to run the sampling algorithm for "long enough".

Recall from Eq. (10) that when the predicted class matches the true class, the density of the confidence prior combined with the categorical likelihood is equal to the density of a cold categorical likelihood. At the same time, the cold likelihood provides a lower bound on the product of the confidence prior with untempered categorical likelihood (because $\hat{y}_y \leq \max_k \hat{y}_k$). To obtain an upper bound on the product, note that conditional on a specific value of the predicted probability for the true class $\hat{y}_y$, the density of the confidence prior is maximized when the rest of the probability mass $(1 - \hat{y}_y)$ is entirely concentrated in a single class. Combined, this yields the following lower and upper bounds:

$$\hat{y}_y^{1/T} \leq p(\hat{\boldsymbol{y}}) \cdot \hat{y}_y \leq \max(\hat{y}_y, 1 - \hat{y}_y)^{1/T-1} \cdot \hat{y}_y. \tag{12}$$

We show that both a cold likelihood and the confidence prior combined with the categorical likelihood converge to the Dirac delta measure by deriving the CDFs of both distributions and showing that the CDFs converge to the CDF of the Dirac delta measure as $T \to 0$.

We can think of the cold likelihood as an (unnormalized) distribution over the predicted probability of the true class. By doing so, we can derive its CDF:

$$\begin{aligned} F_{\text{cold}}(z) &= p(\hat{y} \leq z) \\ &= z^{1+1/T}. \end{aligned} \tag{13}$$

Notice that as $T \to 0$, the cold likelihood CDF converges to the CDF of the Dirac delta measure concentrated at 1:

$$F_\delta(z) = \begin{cases} 0 & z < 1 \\ 1 & z = 1. \end{cases} \tag{14}$$

Similarly, we can derive the CDF for the upper bound on the product of the confidence prior and the untempered categorical likelihood introduced in Eq. (12):

$$F_{\text{up}}(z) = \begin{cases} \dfrac{2^{1/T}\left(T - (1-z)^{1/T}(T+z)\right)}{(2^{1/T}-1)(T+1)} & z \leq 0.5 \\[4mm] \dfrac{2T + \frac{1}{1-2^{1/T}}}{2(T+1)} + \dfrac{(2z)^{1/T+1} - 1}{2(2^{1/T}-1)(T+1)} & z > 0.5 \end{cases} \tag{15}$$

In the limit of $T \to 0$, the upper bound converges to the same distribution as the cold likelihood: the Dirac delta measure. At the same time, recall from Eq. (12) that the cold likelihood provides a *lower* bound for the product of the confidence prior and the categorical likelihood. Therefore, both the lower bound and the upper bound converge to the same distribution as the cold likelihood. In Figure 15, we plot the Wasserstein distance between the cold likelihood and the upper bound. Notice that the Wasserstein distance is exactly zero at $T = 1$ (because the confidence prior is equal to a uniform prior), it peaks around $T \approx 0.1$, and then again converges to zero as $T \to 0$.

Lastly, to verify that the CDFs derived in Eqs. (13) and (15) are correct, we compared them to empirical CDFs obtained using Monte Carlo simulation in Figure 16.

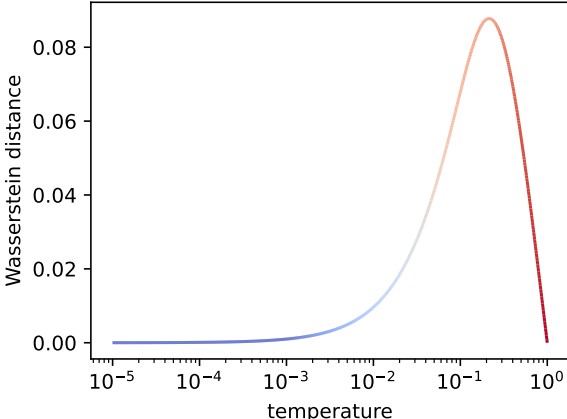

Figure 15: Wasserstein distance between the cold likelihood and the upper bound on the product of confidence prior and untempered categorical likelihood.

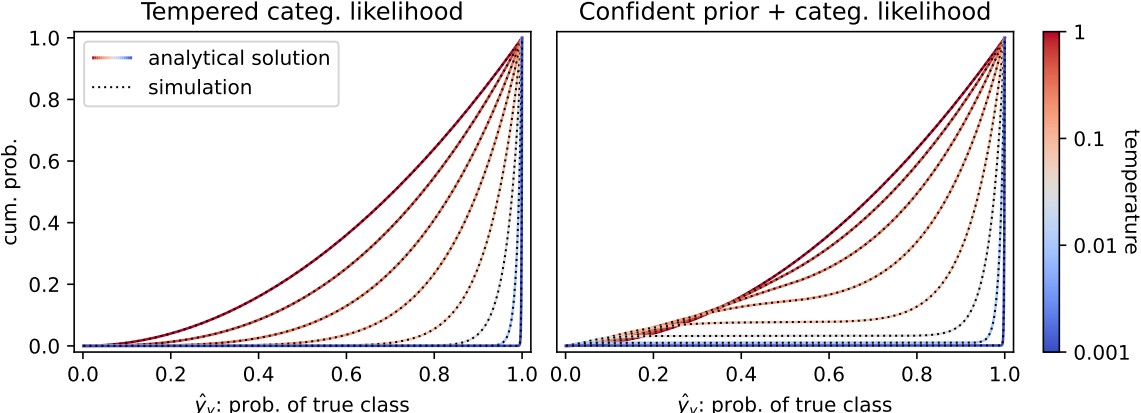

Figure 16: **Left:** CDF of the categorical likelihood tempered to various temperatures (shown by the colorbar on the right). **Right:** CDF of the upper bound on the product of a confident prior (parameterized using temperature) with the untempered categorical likelihood. Notice that the analytical CDFs (solid lines) perfectly match the empirical CDFs obtained using Monte Carlo simulation (dotted lines).

## Appendix F. Priors over outputs

### F.1. Change of variables

Recall from Section 3 that the Dirichlet prior is a distribution over model predictions $p(\hat{\mathbf{y}})$ whereas we ultimately want to sample a distribution over model parameters $p(\boldsymbol{\theta})$, as defined in Eq. (1). So how can we translate the distribution over model predictions to a distribution

over model parameters? Unfortunately, this is not as simple as setting $p(\boldsymbol{\theta}) = p(\hat{\mathbf{y}})$. The issue is that probability *density* is not conserved when transforming random variables; only probability *mass* is conserved. Intuitively, the probability mass of $\boldsymbol{\theta}$ being in some small volume $|\Delta\boldsymbol{\theta}|$ is $p(\boldsymbol{\theta})|\Delta\boldsymbol{\theta}|$, and by applying the same logic to $\hat{\mathbf{y}}$, we get:

$$p(\boldsymbol{\theta})|\Delta\boldsymbol{\theta}| = p(\hat{\mathbf{y}})|\Delta\hat{\mathbf{y}}| \tag{16}$$

$$p(\boldsymbol{\theta}) = p(\hat{\mathbf{y}})\frac{|\Delta\hat{\mathbf{y}}|}{|\Delta\boldsymbol{\theta}|}. \tag{17}$$

When the transformation is invertible, the change in volumes can be computed using the Jacobian determinant. Unfortunately in our case, the mapping from model parameters to model predictions is many-to-one, and therefore the standard Jacobian determinant method does not apply. For a further discussion of how this transform can be approximated, we recommend the work of Qiu et al. (2023). We consider approximating this transformation to be beyond the scope of this paper, although it might be an exciting topic for future research.

We simply note that neither we nor Kapoor et al. (2022) have attempted to compute the ratio of volumes $\frac{|\Delta\hat{\mathbf{y}}|}{|\Delta\boldsymbol{\theta}|}$, i.e. the "change of variables" correction term. Without this correction term, setting the prior distribution over parameters to $p(\boldsymbol{\theta}) = p(\hat{\mathbf{y}})$ does *not* result in the prior predictive distribution following $\hat{\mathbf{y}} \sim p(\hat{\mathbf{y}})$. However, setting $p(\boldsymbol{\theta}) = p(\hat{\mathbf{y}})$ does result in *some* distribution over model parameters, and we simply need to ensure that this distribution is valid (i.e. integrable). When the Dirichlet prior is applied directly over model parameters, its integral diverges, so it is not a valid distribution (we prove this in Appendix F.2).

In contrast, the *DirClip* prior density is bounded by some constant $p(\hat{\mathbf{y}}) \leq B$, so if the DirClip prior is combined with a proper prior over model parameters (e.g. $f(\boldsymbol{\theta}) = \mathcal{N}(0,1)$), the integral of the joint density converges:

$$\int p(\hat{\mathbf{y}})f(\boldsymbol{\theta})\mathrm{d}\boldsymbol{\theta} \leq \int Bf(\boldsymbol{\theta})\mathrm{d}\boldsymbol{\theta} \tag{18}$$

$$\leq B. \tag{19}$$

This implies that the DirClip prior combined with a vague Normal prior corresponds to a proper prior over model parameters.

### F.2. Proof that Dirichlet diverges

For a probability density function to be valid, it must integrate to one. Technically, SGHMC doesn't require the distribution to be normalized, so the distribution only needs to integrate to a *constant*. However, the integral of the Dirichlet prior over model parameters diverges. We prove this by looking at the probability density of a single model parameter, conditional on all other model parameters $p(\theta_1|\theta_2\ldots\theta_N)$. More specifically, we took a ResNet20 trained on CIFAR-10 and fixed the values of all model parameters except a single "bias" parameter in the linear output layer, which we denote $\theta_1$.

Recall from Eq. (3) that the density of the Dirichlet distribution is $\log p(\hat{\mathbf{y}}) \stackrel{c}{=} \sum_{k=1}^{K}(\alpha - 1)\log\hat{y}_k$, which diverges to $\infty$ as one of the predicted probabilities $\hat{y}_k$ approaches zero. By

changing the value of the neural network's bias parameter $\theta_1$, we are directly shifting the value of the output logit corresponding to the first class. Since the neural network is using the softmax activation function, changing the value of a single logit directly changes the predicted probabilities. As $\theta_1 \to \infty$, the predicted probability for the first class approaches 1, while the predicted probability for all other classes approaches 0, causing the Dirichlet probability density to diverge to $\infty$. Conversely, as $\theta_1 \to -\infty$, the predicted probability for the first class approaches 0, thereby also causing the Dirichlet probability density to diverge to $\infty$. We numerically verified this using a ResNet20 trained on CIFAR-10, as shown in Figure 17. However, we note that this behavior generalizes to any neural network using a linear output layer with a bias parameter and the softmax activation function.

In summary, the probability density of $p(\theta_1|\theta_2 \ldots \theta_N)$ diverges to $\infty$ as $\theta_1 \to \pm\infty$. Since the domain of $\theta_1$ is the real numbers, the integral $\int_{\infty}^{-\infty} p(\theta_1|\theta_2 \ldots \theta_N)\mathrm{d}\theta_1$ diverges, meaning that the conditional distribution of the first bias parameter is not a valid probability distribution. This implies that joint distribution over all model parameters is also not valid. Moreover, this divergence applies not only to the Dirichlet prior, but also the Dirichlet posterior that is obtained by combining the Dirichlet prior with a categorical likelihood. We show this is Figure 17.

In practice, these are not just "pedantic" details. The fact that the Dirichlet posterior is not a valid distribution directly implies that it cannot be sampled. If we attempt to sample the Dirichlet posterior using SGHMC, or even optimize it using SGD, the model parameters will diverge. Both we and Kapoor et al. (2022) have observed this behavior.

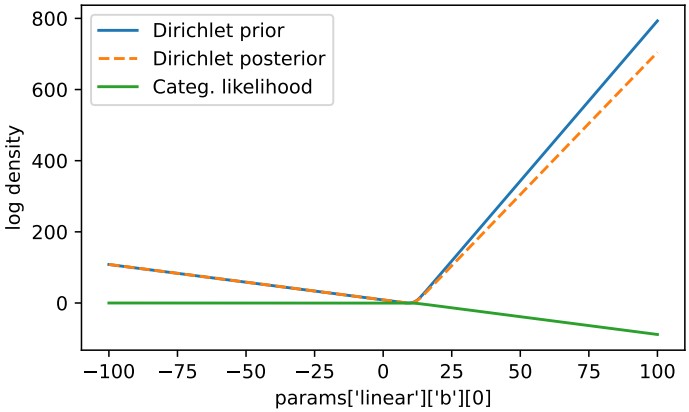

Figure 17: **Dirichlet prior diverges**. We took a ResNet20 trained on CIFAR-10 and fixed the values of all parameters except a single bias parameter in the linear output layer. As we vary the value of this parameter, both the Dirichlet prior and posterior diverge.

## Appendix G. Dirichlet gradient step direction

In this section, we are interested in answering whether a small gradient step along the simplex of a probability distribution over classes will increase or decrease the probability of the true class. First, we derive a general formula for the update of any probability

distribution. Second, we apply this formula to the Dirichlet distribution, discovering the update may decrease the probability of the true class when a small concentration parameter $\alpha$ is used.

Let's denote the predicted probabilities over classes $\hat{\mathbf{y}} = (\hat{y}_1, \hat{y}_2 \ldots \hat{y}_K)$. Since $\hat{\mathbf{y}}$ is only defined on the $(K-1)$-simplex (i.e. class probabilities must sum to 1), it is not possible to perform gradient ascent directly on $\hat{\mathbf{y}}$. Instead, it is more practical to parameterize the predictions using logits $\mathbf{z}$, where $\log \hat{\mathbf{y}} = \mathrm{logsoftmax}(\mathbf{z})$.

Additionally, let's denote the probability density function of interest $f$, so that the log density assigned to any prediction over classes is $\log f(\hat{\mathbf{y}}) = \log f(\mathrm{logsoftmax}(\mathbf{z}))$. A gradient ascent update of logits is therefore $\Delta \mathbf{z} = \epsilon \nabla_{\log f} \mathbf{J}_{\mathrm{logsoftmax}}$, where $\epsilon$ is the learning rate, $\nabla_{\log f}$ is the gradient of the log-PDF w.r.t. $\hat{\mathbf{y}}$ and the Jacobian of the logsoftmax function maps the change in $\hat{\mathbf{y}}$ to a change in $\mathbf{z}$.

Given a change in logits $\Delta \mathbf{z}$, we can linearly approximate the change in class log-probabilities $\Delta \log \hat{\mathbf{y}}$:

$$\begin{aligned} \Delta \log \hat{\mathbf{y}} &= \mathbf{J}_{\mathrm{logsoftmax}} \Delta \mathbf{z} \\ &= \epsilon \mathbf{J}_{\mathrm{logsoftmax}} \nabla_{\log f} \mathbf{J}_{\mathrm{logsoftmax}}. \end{aligned} \tag{20}$$

To understand the update in Eq. (20), we must derive the Jacobian of the logsoftmax function:

$$\mathrm{logsoftmax}(\mathbf{z})_i = z_i - \log \sum_k \exp z_k \tag{21}$$

$$\frac{\partial \mathrm{logsoftmax}(\mathbf{z})_i}{\partial z_j} = \mathbb{1}(i = j) - \hat{y}_j \tag{22}$$

$$\mathbf{J}_{\mathrm{logsoftmax}} = I - \mathbf{1} \otimes \hat{\mathbf{y}}, \tag{23}$$

and the gradient of the Dirichlet log-PDF:

$$\log f(\hat{\mathbf{y}}) = \sum_{k=1}^{K} (\alpha_k - 1) \log \hat{y}_k \tag{24}$$

$$\nabla_{\log f} = \boldsymbol{\alpha} - 1. \tag{25}$$

For convenience, let's denote the gradient of the log-PDF as $\mathbf{g} = \nabla_{\log f}$. Also, let's denote $g^+ = \sum_{k=1}^{K} g_k$ and $\hat{y}^{2+} = \sum_{k=1}^{K} \hat{y}_k^2$. Plugging this into Eq. (20), we get:

$$\begin{aligned} \Delta \log \hat{\mathbf{y}} &= \epsilon \mathbf{J}_{\mathrm{logsoftmax}} (\nabla_{\mathrm{logpdf}} \mathbf{J}_{\mathrm{logsoftmax}}) \tag{26} \\ &= \epsilon (I - \mathbf{1} \otimes \hat{\mathbf{y}}) (\mathbf{g}(I - \mathbf{1} \otimes \hat{\mathbf{y}})) \tag{27} \\ &= \epsilon (I - \mathbf{1} \otimes \hat{\mathbf{y}}) (\mathbf{g} - g^+ \hat{\mathbf{y}}) \tag{28} \\ &= \epsilon (\mathbf{g} - g^+ \hat{\mathbf{y}} - \hat{\mathbf{y}} \cdot \mathbf{g} + g^+ \hat{y}^{2+}). \tag{29} \end{aligned}$$

Gradient ascent increases the probability of the true class iff $\Delta \log \hat{\mathbf{y}}_y > 0$:

$$g_y - g^+ \hat{y}_y - \hat{\mathbf{y}} \cdot \mathbf{g} + g^+ \hat{y}^{2+} > 0. \tag{30}$$

Eq. (30) provides a general condition that must hold for any distribution (and any particular prediction $\hat{\mathbf{y}}$) for the update of the true class to be positive. In Figure 9, we show how the

update depends on both the distribution parameter $\alpha$ and the prediction $\hat{\mathbf{y}}$. To gain more insight into the Dirichlet distribution, we can look at the "worst-case" scenario in terms of $\hat{\mathbf{y}}$. In particular, we are interested in the case where the prior gradient dominates the likelihood gradient, and as a result, the combined gradient decreases the probability of the true class. Observe in Figure 8 that the gradient of the Dirichlet prior grows the closer the prediction gets to a single class. Therefore, the "worst-case" scenario is that $\hat{\mathbf{y}}$ is concentrated around a single class $j \neq y$. By applying this worst-case assumption, we get a minimum value of $\alpha$ that must be used so that the probability update for the true class is positive for all values of $\hat{\mathbf{y}}$:

$$g_y - g^+ \hat{y}_y - \hat{y}_j g_j + g^+ \hat{y}_j^2 > 0 \tag{31}$$

$$g_y - g_j \hat{y}_j + g^+ (\hat{y}_j^2 - \hat{y}_y) > 0 \tag{32}$$

$$g_y - g_j + g^+ \gtrsim 0 \tag{33}$$

$$\alpha - (\alpha - 1) + (K\alpha - K + 1) \gtrsim 0 \tag{34}$$

$$2 + K\alpha - K \gtrsim 0 \tag{35}$$

$$K\alpha \gtrsim K - 2 \tag{36}$$

$$\alpha \gtrsim \frac{K - 2}{K} \tag{37}$$

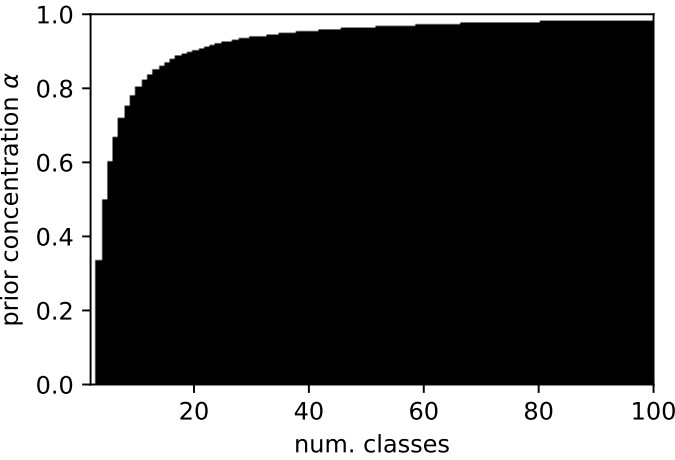

Figure 18: **Phase diagram showing the "critical value" of the Dirichlet concentration parameter $\alpha$.** When $\alpha$ is *above* the critical threshold (i.e. in the white region), then a gradient step on the Dirichlet posterior will always increase the probability of the true class. On the other hand, when $\alpha$ is *below* the critical threshold (i.e. in the black region), gradient steps might *decrease* the probability of the true class, making optimization (or sampling) very challenging.

Notice that the critical value of $\alpha$ in Eq. (37) depends on the number of classes $K$. This relationship is illustrated in Figure 18. For 10 classes, the critical value of $\alpha$ is 0.8, which is consistent with all of our experiments in Section 5. As the number of classes grows, the

critical value of $\alpha$ increases, approaching 1. Note that the increasing $\alpha$ *decreases* the prior concentration; in particular, $\alpha = 1$ corresponds to a *uniform* prior. This is an unfortunate result, meaning that the Dirichlet prior gets increasingly unstable with a growing number of dimensions.

## Appendix H. Implementation details

**Goal.** In order to minimize bias when comparing different posterior distributions, we considered it important to obtain high-fidelity approximations for each posterior. In total, we spent approximately 30 *million* epochs (750 TPU-core-days) to sample all ResNet20 posteriors on CIFAR-10 using SGHMC Chen et al. (2014). It was unfortunately this high computational cost that prevented us from testing more model architectures or datasets.

**Basic setup.** All experiments were implemented in JAX (Bradbury et al., 2018) using TPU v3-8 devices. The typical way to draw SGHMC samples is to generate a single chain where each sample depends on the previous samples. Since BNN posteriors are multimodal (Garipov et al., 2018; Fort et al., 2019; Wilson and Izmailov, 2020), generating autocorrelated samples means that the posterior distribution is explored slowly (Sharma et al., 2023). To eliminate this autocorrelation (and thus achieve more accurate posterior approximations), we instead generated each posterior sample from a different random initialization, completely independently of the other samples.

For most experiments, we followed the learning rate and temperature schedule depicted in Figure 19. Initially, the temperature is zero and the learning rate is high so that the sampler quickly converges to a local mode. Afterward, the temperature is increased to explore the posterior, and the learning rate is decreased to reduce the bias of the sampler. At the end of the cycle, only a single posterior sample is produced. We run this procedure in parallel across TPU cores to generate multiple posterior samples. We provide specific details for each experiment below.

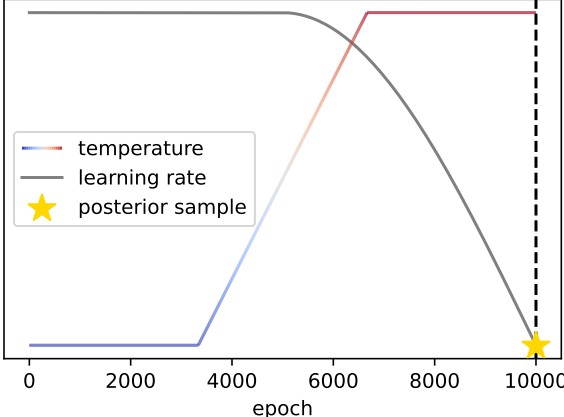

Figure 19: **SGHMC learning rate and temperate schedule**. Temperature starts at zero, increases linearly after $\frac{1}{3}$ epochs and stays at the maximum value after $\frac{2}{3}$ epochs. Learning rate is constant for $\frac{1}{2}$ epochs and then follows a sine schedule to zero.

**Training from scratch.** For the experiments where we compared posterior tempering, DirClip, and NDG against each other (Figs. 3, 4, 7, 10, 11 and 14), a single posterior consists of 8 independent SGHMC samples, each using 10,000 epochs. To obtain error bars, this procedure was repeated three times, therefore leading to 24 posterior samples per model. However, these additional samples were *only* used for uncertainty estimates; the mean accuracy and likelihood only correspond to a posterior consisting of 8 samples. Each posterior used a $\mathcal{N}(0, 0.1^2)$ prior over model parameters and a learning rate of $10^{-4}$. The learning rate was intentionally very low and the number of epochs very high so that each posterior sample would converge to the MCMC stationary distribution.

**Are 8 posterior samples enough?** Across all of our experiments, we used 8–16 *independent* SGHMC samples to approximate each posterior distribution. This is a distinct approach to prior works which generated long correlated SGHMC chains. We chose this approach to eliminate any autocorrelation between posterior samples but also to obtain an algorithm that parallelizes more easily across TPU cores (allowing us to use relatively small batch sizes without incurring significant overhead). Using only 8 independent samples, and the same augmentation strategy as Wenzel et al. (2020), we matched or slightly exceeded the test accuracy of ResNet20 trained on CIFAR-10 of prior works (Wenzel et al., 2020; Kapoor et al., 2022; Fortuin et al., 2021). In Figure 20, we show that we could further improve the test accuracy (and especially the test likelihood) by using more posterior samples.

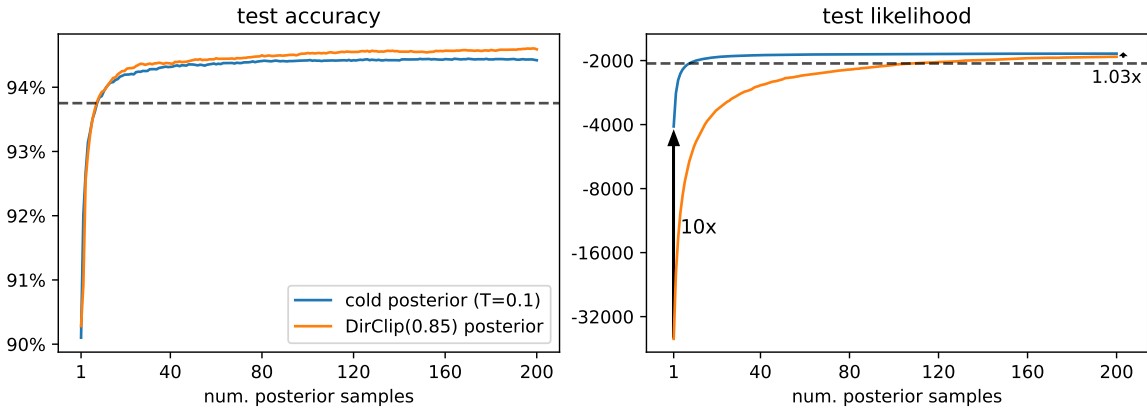

Figure 20: **Posterior size**. Across all of our ResNet20 experiments, we used completely independent SGHMC posterior samples. The dashed line shows the test accuracy and likelihood for a cold posterior consisting of 8 samples. Observe that 8 samples are enough for the posterior accuracy of both a cold posterior and a DirClip posterior to converge within 1%. However, the test likelihood of the DirClip posterior takes much longer to converge.

**Fine-tuning.** In Figures 4, 7 and 10, some of the DirClip models were initialized from a pretrained model. More specifically, this means that the SGHMC sampler was initialized from a model with 100% training accuracy, obtained using SGD. Given the unstable dynamics of fine-tuning, we ran the SGHMC sampler using a very low fixed learning rate and a fixed temperature $T = 1$. The DirClip model with a clipping value of $-50$ used

a learning rate of $10^{-7}$ and 10,000 epochs per sample. The DirClip models with a clipping value of $-10$ and $-20$ used an even lower learning rate $(3 \cdot 10^{-8})$ and an increased 50,000 epochs per sample, to ensure better numerical stability for small $\alpha$.

**Confidence of a Normal prior.** For the experiments shown in Figures 5 and 6, we sampled 15 different priors scales across 16 temperatures, leading to 240 unique posterior distributions. Given the large number of different distributions, we opted for the lowest sampling fidelity for this experiment. A single posterior consists of 16 independent SGHMC samples, each using 1,000 epochs. The learning rate was tuned for each prior scale to maximize the test accuracy of a single posterior sample. Once the learning rate was tuned, the 16 posterior samples were drawn using a different random seed, and the previous samples were discarded to avoid any data leakage.

**2D toy example.** Figure 1 shows the only experiment that we ran on a CPU rather than TPUs. We used a fully-connected neural network with 5 hidden layers of 10 neurons, using the ReLU activation function. We used HMC Neal et al. (2011) with 10,000 leapfrog steps and 1,500 samples per posterior. Both chains achieved an $\hat{R}$ metric (Brooks and Gelman, 1998) of 1.00 in function space and $\leq 1.05$ in parameter space. We used a DirClip clipping value of $-50$.

**Training accuracy.** Note that in Figures 3 and 4, the training accuracy was evaluated on posterior *samples* rather than the posterior predictive distribution—the goal of these plots was to explicitly visualize the behavior of posterior samples. In contrast, test accuracy was measured using an ensemble of 8 posterior samples, to provide a direct measure of the posterior performance.

**Data augmentation.** We used exactly the same data augmentation strategy as Wenzel et al. (2020): left/right flip, border-pad, and random crop.

