# OpenReview forum: "Can a Confident Prior Replace a Cold Posterior?"
_approximateinference.org/AABI/2024/Symposium — AABI 2024_

### Official Review · Reviewer_XrYG · 2024-04-19
**Clipping Dirichlet Prior to help address cold-posterior effects in bayesian neural netwoks.**

**Rating:** 8
**Confidence:** 4

**Review:**

This paper discusses how we can control complexity in the decision boundaries for Bayesian neural networks through a prior distribution over classification outputs. It is common in the BNN literature to deal with this through use of a cold-posterior, though has theoretical limitations and points to a misspecification within our model. The Dirclip prior distribution is introduced here, which shows how it can target a valid Bayesian posterior and achieve similar predictive performance to that of the cold-posterior effect.

The paper builds upon the work of the Noisy-Dirichely-Gaussian (NDG) prior used by (Kapoor etal 2022) which combines a valid prior with a quadratic likelihood term, and provides evidence that the performance obtained from NDG relies on both these used in conjunction. This paper follows a similar approach, but shows that through a simpler clipping mechanism can insure that the resulting PDF is bounded and that divergences can be avoided. Experimental results on simple toy datasets visualise how the Dirclip prior can either temper or sharpen a decision boundary for a simple regression task, and that in a classification setting can result in improved predictive performance similar to that of when a cold-posterior is used.

The paper also clearly acknowledges limitations in this approach which is wonderful to see. For the classification setting, they show that if the Dirichlet component of Dirclip is not specified appropriately, then the model can fail to yield meaningful results. They also similarly acknowledge that this type of result is often seen in a simple optimisation setting, such that simply setting a learning rate or momentum parameter in SGD incorrectly can yield similar results.

The paper provides thorough and detail descriptions of experimentation procedure and derivation of theoritical results in the supplementary material.

I am curious about how some of this work may link to that of Aitchison 2021, which poses that the effect of a cold-posterior may be largely due to an incorrectly specified likelihood function. Given that the the Dirclip is more placing a prior closer to the likelihood space, I wonder if there is a link between these two papers that might be worth exploring? Also curious about  the potential for learning of some of the hyperparameters used within Dirclip. Could a hyperprior help with exploring suitable $\alpha$ parameters, or with the clipping parameter $v$? This may complicate inference, but perhaps a hyperprior that is vague enough to allow for sufficient exploration but constrained enough to avoid problematic regions may make inference easier practically speaking.

Overall this is a great paper showing meaningful new results addressing the common issue of predictive performance in BNNs. I believe this paper would be a great contribution to AABI symposium and recommend acceptance.


## References

  Aitchison, Laurence. "A statistical theory of cold posteriors in deep neural networks.

---

### Official Review · Reviewer_9Fon · 2024-04-24
**Paper discusses issue in noisy Dirichlet approximation of BNN posterior and proposes a simple (hacky) fix**

**Rating:** 5
**Confidence:** 3

**Review:**

The paper examines the noisy Dirichlet (Gaussian) prior over a Bayesian Neural Network described in (Kapoor et. al, 2022) and proposes a simple modification to the Dirichlet density addition to the standard BNN prior to ensure that the new (noisy Dirichlet) prior is a proper prior. Their method simply clips the log probabilities in the Dirichlet log density to a user selected value so as to prevent the log density becoming infinite. They contrast this method with the Gaussian approximation to the posterior used in (Kapoor et. al, 2022) which they interpret as a Gaussian prior multiplied with a quadratic likelihood. Their results indicate that lower clip values achieve better test accuracy for various concentration values of the Dirichlet density.

The paper took more time to understand than expected due to vague notation and language (e.g. $p(\theta) = p(\hat{y})$ in Section 3 and Appendix F.1 was confusing due to the omission of $\hat{y}$'s dependence on $\theta$ (and $x$) and the phrase "Dirichlet prior as a prior over parameters" in Section 3 can be easily misinterpreted to mean a Dirichlet distribution on the BNN weights/biases) and the reviewer needed to consult (Kapoor et. al, 2022) to understand some of the vague notation given in the paper. The idea of clipping the log probabilities in the Dirichlet density seems to be a minor contribution and a bit of a hack that would be better suited to a short note/blogpost. However, the analysis of their DirClip results (e.g. Appendix C) and the NDG approximation (Appendix D) offers deeper insights into the two approaches and how to set the model hyperparameters.


References:
Sanyam Kapoor, Wesley Maddox, Pavel Izmailov, and Andrew Gordon Wilson. On uncer- tainty, tempering, and data augmentation in bayesian classification. In Alice H. Oh, Alekh Agarwal, Danielle Belgrave, and Kyunghyun Cho, editors, Advances in Neural Informa- tion Processing Systems, 2022.

---

### Official Review · Reviewer_XxaX · 2024-04-26
**Paper Review**

**Rating:** 5
**Confidence:** 3

**Review:**

# Overview

In this paper, the authors examine the cold posterior effect.
In particular, they consider an existing method, the Dirichlet prior over outputs of Kapoor et al. (2022), as a principled Bayesian fix for the cold posterior effect.
They propose a modification, the DirClip prior, which they argue resolves a pathological behaviour of the Dirichlet prior.

# Quality & clarity
Overall, the quality and clarity of the presentation of the paper could be improved.
I found that the exposition of the Dirichlet prior in the paper is not clear in my opinion, and I had to consult Kapoor et al. (2022) to get an understanding of what the existing method is.
I expand on this point further in the comments below (see Section 3 (Dirichlet prior) could be clearer).
Concretely, I would suggest bringing in some of the explanation from Kapoor et al. (2022) into the paper and modifying the notation to stress that the Dirichlet and DirClip priors depend on the model parameters $w$ implicitly via the model predictions $\\hat{y}$ as well as on the inputs $x.$
I thought that the experiments presented in the paper were decent.
Overall, given the relative lack of clarity and considerations on mathematical accuracy, I would say the overall quality of the paper is somewhat below average and could be improved.

# Originality & Significance
The topic examined by the paper is an important one, and the fix proposed is relatively simple (which I consider a benefit).
However, it is not totally clear to me that the proposed approach fixes the issues of the Dirichlet prior discussed in the paper.
In particular, it does not seem to me that the DirClip prior is integrable (i.e. I don't think it integrates to 1).
I expand on this in the comments below (see comment on proofs (2)).

# Comments and points for improvement

__Comment on aleatoric uncertainty in classification:__
The following comment made by the authors did not fully make sense to me.

> In contrast, in a classification setting, we are forced to use the categorical likelihood, which has no tunable parameter to control the level of aleatoric uncertainty.

Maybe I misunderstood this, but I don’t think this is entirely right.
For example, one can use a positive scalar multiplier to scale the logits entering the softmax distribution, making it more peaked or more uniform.
For example if the logits that are fed as input to the softmax are $z = (z_1, \dots, z_N),$ then one could set the output of the network to be $\\texttt{softmax}(\\alpha z)$ instead of $\texttt{softmax}(z),$ where $\alpha > 0$ is a tunable parameter.
As $\\alpha \\to 0,$ we have $\\texttt{softmax}(\\alpha z) \\to 1/N,$ i.e. the aleatoric uncertainty is maximised.
As $\\alpha \\to \infty,$ we have $\\texttt{softmax}(\\alpha z) \\to \\mathbb{1}_{i = \\text{argmax} (z)},$ i.e. there is no aleatoric uncertainty.
Have the authors considered performing an experiment involving a simple method like this?

__Section 3 (Dirichlet prior) could be clearer:__
I think the introduction of the standard Dirichlet prior method could have been clearer.
For example, from eq. (3) it is not clear how the Dirichlet prior relates to model parameters.
After reading through the reference provided (Kapoor et al. 2022), I have the impression that $\\hat{y}$ depend on the inputs $x,$ which are however not written out.
If this is the case, this prior is an input-dependent prior and should be written as $p(\\theta | x)$ instead of $p(\\theta).$
Perhaps it is worth clarifying this here (and in the manuscript).
Generally I found the notation in Kapoor et al. (2022) to be clearer than the one provided here, so perhaps some elements of their presentation of background material and notation can be borrowed here.

__Comment on proofs (1):__
In appendix F.2, the authors argue that the Dirichlet prior diverges as any one of the predicted probabilities $\\hat{y}_k$ approaches 0, which can happen if one of the network parameters, say the corresponding bias parameter $\\theta_k$ goes to $\\infty.$
Looking at the Dirichlet prior log-pdf expression (end of page 22), isn't this true only if the concentration parameter is less than 1, i.e. $\\alpha < 1$?
Otherwise, it seems that the log-pdf expression diverges to $- \\infty$ which does not seem to be a problem to me.
Am I missing anything here?
If not, it would be good for the authors to clarify this in the main text and appendix.
Further, isn't this instability fixed if one includes a prior over the weights whose log-pdf decays at a rate faster than linear (e.g. a Gaussian)?

__Comment on proofs (2):__
Related to the above, the authors make the case that the Dirichlet prior is not a valid prior because it does not integrate to 1.
However, it's not clear to me that the proposed fix, DirClip, is a valid prior over parameters either.
It's not clear to me that it integrates to 1 and, in fact, except if I have misunderstood the DirClip prior, I don't think it does. In particular, looking at eq. (4), we see that the log probability is actually bounded below by $K (\\alpha - 1) v,$ where $K$ is the number of classes, $\\alpha$ is the concentration parameter and $v$ is the clipping threshold of DirClip.
Therefore, the probability density that is implied over $w$ would be bounded below by the constant $e^{K (\\alpha - 1) v}.$
Integrating this over the parameter space ($\\hat{y}$ depends on $w$ and $x$) which is unbounded, we would get $\\infty.$
I could perhaps be missing something here but clarification on this in the manuscript would be good.

---

### Official Review · Reviewer_oGia · 2024-05-02
**interesting work!**

**Rating:** 7
**Confidence:** 3

**Review:**

The paper introduces a proper prior that can achieve high performance (accuracy) close to using a cold posterior on BNNs for classification.
It carefully studies the effects of different components introduced in the previous work on how they helps with the CPE and demonstrated that the clipped Dirichlet prior can be used to control the aleatoric uncertainty without introducing any numerical issues during training, hence bringing the gap between models with cold posterior with only changes in prior.

## quality
The method and analysis done in the paper is sound and the experiment setup is valid.
The results are well discussed.
One question I have is that whether or not (4) is normalized?

## clarity
The paper is well-written and easy to follow.
The end of section 2.2 is a neat summary of previous work.

## originality
The new prior used in the context of CPE and analysis&findings are novel

## significance
The paper is an interesting read for the community as to better understand the source of improvement to the CPE from different methods and demonstrate a plausible, "proper Bayesian" way to tackle the problem; it could motivate more studies on this direction.

---

### Meta-Review · Area_Chair_fBoT · 2024-05-24

**Recommendation:** Accept (Poster)
**Confidence:** 3

**Metareview:**

This paper proposes a simple prior for BNN classifiers that is able to perform as well as the "cold posterior" that generally tends to outperform common BNN priors. The reviews for this paper were mixed (with two marginal rejects and two fairly strong accepts). Many of the criticisms stem from clarity issues, which could be addressed for the camera-ready. There were also some concerns about the soundness of proofs. However, given that this paper tackles an interesting topic and also provides experimental evidence showing the prior performs well in practice, I recommend acceptance.

---

### Decision · Program_Chairs · 2024-05-27

Accept